# Aquaphotomic, E-Nose and Electrolyte Leakage to Monitor Quality Changes during the Storage of Ready-to-Eat Rocket

**DOI:** 10.3390/molecules27072252

**Published:** 2022-03-30

**Authors:** Laura Marinoni, Marina Buccheri, Giulia Bianchi, Tiziana M. P. Cattaneo

**Affiliations:** 1Research Centre for Engineering and Agro-Food Processing, Council for Agricultural Research and Economics, 20133 Milano, Italy; marina.buccheri@crea.gov.it (M.B.); giulia.bianchi@crea.gov.it (G.B.); tiziana.cattaneo@crea.gov.it (T.M.P.C.); 2Department of Agricultural and Forestry Sciences (DAFNE), Tuscia University, 01100 Viterbo, Italy

**Keywords:** storage, ready-to-eat rocket, water, NIR, E-nose, electrolyte leakage, Aquaphotomics

## Abstract

The consumption of ready-to-eat (RTE) leafy vegetables has increased rapidly due to changes in consumer diet. RTE products are perceived as fresh, high-quality, and health-promoting. The monitoring of the RTE quality is crucial in relation to safety issues. This study aimed to evaluate the maintenance of RTE rocket salad freshness packed under modified atmospheres. A portable E-nose, the electrolyte leakage test (which measures the index of leaf damage—I_LD_), and NIR spectroscopy and Aquaphotomics were employed. Two trials were carried out, using the following gas mixtures: (A) atmospheric air (21% O_2_, 78% N_2_); (B) 30% O_2_, 70% N_2_; (C) 10% CO_2_, 5% O_2_, 85% N_2_. Samples were stored at 4 °C and analyzed at 0, 1, 4, 7, 11, and 13 days. ANOVA, PCA, PLS were applied for data processing. E-nose and I_LD_ results identified the B atmosphere as the best for maintaining product freshness. NIR spectroscopy was able to group the samples according to the storage time. Aquaphotomics proved to be able to detect changes in the water structure during storage. These preliminary data showed a good agreement NIR/I_LD_ suggesting the use of NIR for non-destructive monitoring of the damage to the plant membranes of RTE rocket salad.

## 1. Introduction

The consumption of ready-to-eat (RTE) leafy vegetables has increased rapidly due to changes that have occurred in the dietary pattern of the consumer. RTE products are perceived as natural, fresh, convenient, high-quality, and health-promoting [1,2]. Rocket (*Eruca sativa* Mill.) salad is principally consumed as an RTE product, either alone or mixed with other vegetables [3]. It shows different characteristics from other types of salad, as it belongs to the Brassicaceae family. For this reason, it is a source of compounds with biological activity beneficial for human health, such as glucosinolates. Glucosinolates are then hydrolyzed by the enzyme myrosinase with the release of isothiocyanates, which are compounds with antimicrobial activity [4]. Because of its high content of bioactive compounds, different health-promoting functions are attributed to rocket plants [5], but, as with most leafy vegetables, rocket leaves have very high metabolic activity [6], which limits their shelf-life. During storage, the high respiration rate and ethylene production lead to leaf yellowing due to chlorophyll degradation, loss of turgidity, off-flavor development, and general deterioration [7].

Many postharvest technologies, such as refrigeration and modified atmosphere packaging (MAP), are successfully employed to delay the senescence of packed vegetables [8]. These techniques are determinants in preserving visual quality and microbial safety of minimally processed fruit and vegetables during the supply chain [9]. MAP is effective in prolonging the shelf-life of fresh-cut produce by modifying the ratios between gases within the packaging. Successful applications of MAP (with low O_2_ and high CO_2_ levels) to minimally processed fruits and vegetables have been extensively reported in the literature [2], since low O_2_ concentration causes a decrease in respiration, this inhibits the growth of postharvest pathogens, and slows down the deterioration rate. However, very low levels of O_2_ may induce anaerobic fermentation with the corresponding accumulation of unpleasant odors, undesirable tastes, and tissue damage [2,8,10]. Moreover, the presence of a very high CO_2_ concentration (25%) in the storage atmosphere has been found to be deleterious for fresh-cut artichokes, while only slight beneficial effects were observed at lower concentrations (5 and 15%) [11].

The monitoring of the quality of RTE vegetables is crucial, especially in relation to safety issues [12]. It is essential to evaluate the product quality both during storage and at the end of its shelf-life. 

Several techniques are used to monitor the quality of fresh-cut fruit and vegetables during shelf-life, such as the evaluation of texture, respiration rate, pH, microbiological and sensory quality, nutritional and antioxidants status [13]. Among them, the electrolyte leakage test measures the cell membrane permeability, resulting in an index of leaf damage. This method is widely used in the investigation of various stress conditions, such as quality changes during shelf-life in different ready-to-eat products [14,15,16], including rocket salad [3,17]. However, all these methods are generally expensive and time-consuming, require the destruction of the sample, and are not suited to automation. 

Among the non-destructive techniques, the electronic nose (E-nose) is a fast and reliable method to evaluate the volatile fingerprint in the headspace of food. According to Gardner and Bartlett [18], the electronic nose is “an instrument which comprises an array of electronic chemical sensors with partial specificity and an appropriate pattern-recognition system, capable of recognizing simple or complex odors”. The sensors respond to the whole set of headspace volatiles, giving an idea of the metabolic changes that take place in the headspace of food [19]. More than 70 volatiles have been identified in the headspace of fresh rucola, 20 of which contribute to the leaf aroma [20]. Some typical volatile compounds found in RTE rocket salad are dimethyl sulfide, dimethyl disulfide, dimethyl sulfoxide, and furans derivatives [20,21].

Near-infrared (NIR) spectroscopy, in combination with chemometrics and Aquaphotomics, represents a powerful, rapid, and non-destructive analytical tool to monitor the quality of packaged foods by evaluating the changes occurring during the storage [22].

Aquaphotomics is a recent scientific discipline based on NIR measurements and multivariate spectral analysis that investigates the water–light interactions in biological systems. This approach exploits the fact that changes in the water matrix reflect, as would a mirror, the molecules the water surrounds [23,24]. Aquaphotomics is based on the high sensitivity of water’s hydrogen bonds that reflect any change in the aqueous system highlighting perturbations that can be observed, measured, analyzed, and interpreted [24]. According to this approach, the NIR spectra acquired in living systems under various perturbations (temperature, ion concentrations, oxidative stress, illumination, disease, and damage) are characterized by 12 water absorption ranges (6–20 nm width each) in the spectral region of the first overtone of water (1300–1600 nm). Such spectral ranges have been called Water Matrix Coordinates (WAMACs) and labelled Ci, i = 1–12. Within the WAMACs, specific water absorbance bands are related to specific water molecular conformations (water species and water molecular structures) [24]. When a perturbation produces changes at specific water absorbance bands, and when this is determined consistently and repeatedly throughout the Aquaphotomics analysis, these water absorbance bands (WABs) are considered ‘activated’ by the respective perturbation. The selected WABs are plotted in spider charts, named ‘aquagrams’, which depict the Water Absorbance Spectral Patterns (WASPs) [24].

This work aimed to evaluate and monitor the changes occurring during the storage of ready-to-eat rocket salad packed under modified atmospheres. The electrolyte leakage test and rapid and non-destructive techniques, such as NIR spectroscopy coupled to Aquaphotomics and a Portable Electronic Nose, were employed. 

## 2. Results and Discussion

### 2.1. E-Nose Analyses

In a previous preliminary work [25], the evaluation of the storage of the fresh-cut spring rocket (Trial 1) was carried out only through two non-destructive techniques, electronic nose and NIR spectroscopy. 

The E-nose was applied in order to evaluate the evolution of the volatile compounds profile of rocket salad during storage. Preliminary results of Trial 1, only summarized here, highlighted the great influence of three broad-range sensors, W5S, W1S, and W2S, on the characterization of the samples. Such sensors are indicated as sensitive towards a wide range of compounds; however, they possess some selectivity towards specific compounds; respectively, methane, and alcohols for W1S and W2S. These compounds, indeed, indicate anaerobic conditions or fermentation reactions likely characterizing the late storage stages, and the related signals mainly characterized the last checkpoints of all the samples. In particular, C samples showed a peculiar pattern, with an important impact of the three sensors at all sampling points. The E-nose profile of treatment C underwent major changes during the storage, while treatments A and B showed more constant sensor values. Profile B reflected minimal variations occurring inside the rocket bag, suggesting that the B atmosphere was the best in maintaining the product’s initial conditions [25].

In the second experiment carried out on second-cut autumn rocket (Trial 2) and fully reported here, the three broad-range sensors were confirmed as the predominant ones in the characterization of the samples. Figure 1 shows the biplot of PCA illustrating the mutual relationships between samples and sensors.

The three broad-range sensors were responsible for the positioning in the multivariate space of the longer-stored samples. W1S and W2S greatly described the sample C11 and, to a lesser extent, C7 and A13. W5S mainly characterized the C13 sample. The other sensors, sensitive to aromatic compounds, alkanes, terpenes, and organic sulfur compounds, were located close to the axes intersection and described the short-term stored samples. These findings are in accordance with Mastrandrea et al. [26], who studied the changes in volatile compounds responsible for flavor in wild rocket stored in MAP conditions. They found volatile and aromatic compounds, including sulfur, C6, and C5 compounds, acetaldehyde, isothiocyanate, and thiocyanate derivatives. These compounds are responsible for the typical odor and flavor notes of fresh rocket [26]. Interestingly, A11, B11, and B13 samples clustered close to the fresher samples, indicating that these two active atmospheres played a crucial role in limiting metabolic changes inside the bags during the storage.

In Trial 2, E-nose profiles differed among the treatments (Figure 2a). Samples A and B showed lower sensor values compared to treatment C, which was characterized, in particular, by very high W5S sensor values. 

Figure 2b reports in detail the trend of the three broad-range sensors. Treatment A showed constant values until t = 11 and then a marked significant increase at t = 13 of all sensors. Treatment B showed significant differences on days 11 and 13 for the W1S and W2S sensors and only at time t = 13 for the W5S. Sample C showed significant differences for all sampling points with maximum values at the last checkpoint, indicating a possible accumulation of a wide range of compounds, such as methane and alcohols, in the package atmosphere. The fact that the three sensors mainly described the samples of treatment C, which is characterized by low oxygen, agrees with the literature data [10,27,28,29]. Indeed, alcohols, such as ethanol and methanol, have been reported to be released under restricted O_2_ conditions [10,27]. Ethanol, for example, is considered a major fermentative metabolite, while methanol is reported to come from enzymatic degradation of pectin by pectin methyl esterase [28]. It can be assumed that in the C treatment, the lack of O_2_ for aerobic respiration led to a switch to anaerobic respiration, with a release of alcohols. Increases in fermentative volatiles such as ethanol are reported to negatively affect the sensory properties of the product [30,31,32]. Indeed, Allende et al. [29] reported that products with high respiration rates require high levels of O_2_ in the package to maintain their quality.

### 2.2. Electrolyte Leakage Analyses

The present work also shows the results, of both trials, of the electrolyte leakage test.

The index of leaf damage (I_LD_), measured by an electrolyte leakage test, can be considered a good indicator of the product’s freshness [33]. In the first trial (spring mowing, Figure 3a), I_LD_ remained more or less constant in all treatments until day 4. On day 7, it increased significantly only in C treatment (LSD > 0.61). In treatments A and B, the increase in I_LD_ took place only after 10 days. In any case, treatments A and B showed lower I_LD_ values than treatment C until the end of the storage (13 days).

In the second trial (autumn mowing, Figure 3b), the leaves’ behavior was slightly different. All the treatments showed an I_LD_ value far higher than in the first trial. I_LD_ values were constant until day 7, after which all the values increased almost exponentially. Moreover, in this case, the C treatment showed the worst performance reaching, at the end of the storage, the highest I_LD_ value (LSD > 2.2).

**Figure 3 molecules-27-02252-f003:**
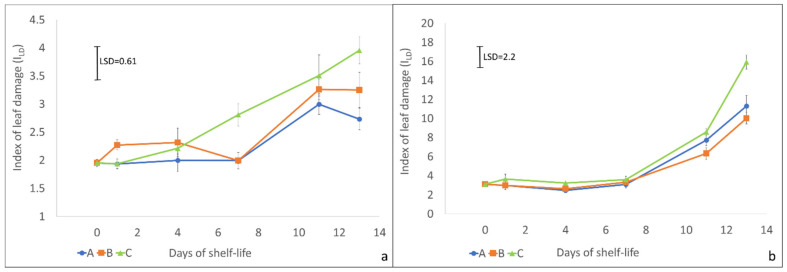
Index of leaf damage (I_LD_) of rocket salad samples of trial 1 (**a**) and 2 (**b**) ± standard error. The result of the LSD post-hoc test is reported in each graph (*p* < 0.05).

Considering both trials, the low I_LD_ values of the samples in the first days agree with the findings of Luca et al. [16], who reported a low index of leaf damage until day 8 of shelf-life in wild rocket leaves. The higher value of treatment C at the end of the storage period indicates that a modified atmosphere enriched with CO_2_ can negatively affect rocket leaves, inducing faster deterioration of the samples. Mastrandrea et al. [34] obtained similar results on rocket leaves under passive modified atmosphere conditions: high CO_2_ and low oxygen concentration increased off-odors production and decreased shelf-life.

The different behavior of rocket leaves (higher deterioration in the second trial) can be due to the two different seasons when the mowing took place. Environmental conditions, including temperature, solar radiation, and rainfall, have an important influence on the quality of brassica leafy species [35,36]. Furthermore, rocket is capable of regrowing after mowing, giving under optimal pedo-climatic conditions up to 5–6 cropping in a season [37]. Hall et al. [38] reported that the harvest season and the mowing number affect the leaf water content, with late cuttings and summer cuttings having higher dry weight (lower water content). In the two experiments, two rocket harvests were analyzed— the first cut in the early spring and the last cut in the early autumn. The different harvest season and the cutting number could have affected the leaves’ composition, and especially their water content [36,38,39]. For this reason, leaves from plants of the first trial (April) were probably more resistant to deterioration than those of the second trial (October).

### 2.3. NIR Spectroscopy and Aquaphotomics

NIR data of Trial 1, as well as those of E-nose, have recently been presented as a poster [25]. In the present paper, these are reported in more detail and compared with those of Trial 2 and with the electrolyte leakage data. 

Figure 4 shows the averaged raw NIR spectra of the packed rocket salad samples of Trial 1 (a) and 2 (b). The raw spectra showed broad and overlapped bands related to water absorptions at 960–990 (second overtone -OH stretching), 1150–1170 (combination of the first overtone -OH stretching and OH bending), and 1430–1440 nm (first overtone -OH stretching) [24]. The spectra of leafy vegetables originate from the interaction of the electromagnetic radiation with the compounds that absorb it (chlorophyll, carotenoids, water, cellulose and hemicelluloses, starch, lignin, proteins) and from reflection and internal scattering phenomena of the non-absorbed radiation [40]. The leaf can be modeled as a stack of four layers, each of which owns different physical characteristics and optical properties. The light interacting with the leaf can be subjected to phenomena of specular reflection, absorption, transmission, and scattering [40,41,42]. The difference in vegetable surface, thickness, and structure, as occurs for leaves and petioles, determines a different reflection of the electromagnetic radiation [40]. Furthermore, the growing season is recognized to have an influence on the chemical-physical characteristics of the leaves, such as the amount of dry matter, the leaf area surface, and the thickness [43]. Bonasia et al. [44] reported higher dry matter concentration, higher specific leaf area, and more thickened leaves for rocket salad grown in winter-spring compared to the autumn-winter product. The differences in the sample composition caused by the growing season could have affected the raw spectra of Trial 2, which were more heterogeneous and dispersed than in the first trial. Furthermore, rocket leaves from Trial 2 (October mowing) were more variable in shape and dimensions, while the petioles were thicker than those from Trial 1 (April mowing). This distinct leaf morphology probably caused a change in the arrangement of the layers of leaves and petioles inside the package that gave rise to differences in the optical phenomena in the two systems, resulting in different spectra. 

The application of pre-treatments made the peaks narrower and more defined, allowing their easier identification. Figure 4c,d show the detail of the second derivative spectra truncated to 1300–1600 nm in order to examine the first overtone of the water. The presence of wavelength shifts in the analyzed range supported the hypothesis that Aquaphotomics could be helpful to discriminate between the different treatments.

**Figure 4 molecules-27-02252-f004:**
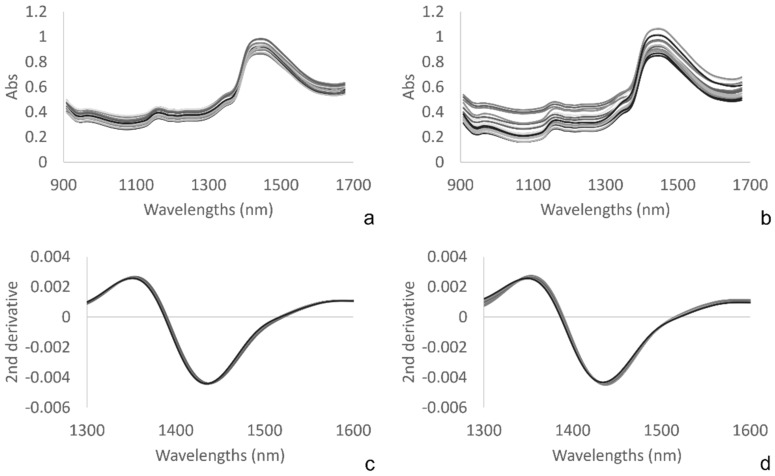
Averaged raw NIR spectra of the packed rocket salad samples of Trial 1 (**a**) and 2 (**b**) and detail of the second derivative spectra truncated to 1300–1600 nm of Trial 1(**c**) and 2 (**d**).

#### 2.3.1. Trial 1

To highlight and enhance the small differences found in the second derivative of the spectra, the normalization was performed on Trial 1 spectra according to Tsenkova et al. [24] (Figure 5).

The graph allowed the identification of the wavelengths most involved in the characterization of the different samples. In order to identify further characterizing wavelengths, the PCA and PLS models were calculated on the pre-treated spectra. 

The explorative PCA explained 98% of the total variance showing samples grouped according to the storage time (Figure 6): (i) t = 0; (ii) t = 1 samples; (iii) t = 4 samples; (iv) t = 7 to t = 13 samples. Interestingly, sample B at 7 days located together with t = 4 samples. This suggested that the characteristics of the biosystem of the 4 days samples were also maintained for 7 days rocket samples stored in atmosphere B.

According to the loadings plot (Figure 6b), the separation along PC1 between fresh and longer-stored samples was mainly based on a wavelength of 1373 nm, while the separation along PC2 was due to 1453 nm. According to Tsenkova et al. [24], these wavelengths correspond to the absorbance of the free OH stretch and of strongly hydrogen-bonded water, respectively.

A preliminary PLS model was built up with pre-treated spectra and the I_LD_ index. Figure 7 shows the scatter plot of the model (Figure 7a) together with the factor loadings (Figure 7b).

The model showed an R-value in calibration of 0.92 and an R-value in cross-validation of 0.83, with a standard error in cross-validation of 0.30%. The good performance of the preliminary predictive model indicated some relationship between the NIR spectra and the plant membrane damage index. The NIR spectrum of a leaf is indeed strongly influenced by its structure, thickness, and state of hydration, which in turn is an index of plant stress [45]. Aday [46] also found agreement between FT-NIR spectroscopy and electrolyte leakage data on mushrooms treated with electrolyzed water in combination with passive atmosphere packaging. 

Based on the common WABs found in normalized spectra and in the PCA and PLS loadings plots, 12 wavelengths inside the WAMACs ranges were selected for the construction of aquagrams. Each wavelength corresponds to the absorption of specific water molecular species. The wavelengths are listed and described in Table 1.

The aquagrams of Trial 1 highlighted differences among the samples, giving rise to different WASPs for each sampling time and for each treatment (Figure 8).

Treatments A and C of the Trial 1 showed a shift from left to right up to the fourth day; then, the graph went back to the left. For treatment B, the inversion occurred on the seventh day. This behavior indicates a prevalence of hydrogen-bonded water species and strongly-bound water [47] in the freshly cut rocket. With the progress of the storage, a situation is reached in which free water or weakly hydrogen-bonded water prevails [47]. This point could indicate the loss of freshness of the product. From these premises, it can be assumed that Aquaphotomics estimated the first loss of freshness after 4 days from packaging for the A and C treatments, and after 7 days for the B treatment. This suggested that treatment B was the best at preserving the freshness of rocket salad, in agreement with the E-nose findings.

#### 2.3.2. Trial 2

Based on the characteristics of the second sampling (autumn rocket, October mowing, presence of larger leaves and long petioles), the adequacy of the Aquaphotomics model was verified. The data processing procedure was repeated as for the first trial.

PCA is reported in Figure 9. 

Again, there was a good separation of the samples according to the days of storage along PC1, which accounted for 98% of the variance (Figure 9a). The group of samples from day 4 was separated from the others along PC2. Interestingly, the B11 sample ranked alongside the first checkpoints, while the C7 sample was superimposed on t = 11/13 samples. 

According to the loadings plot (Figure 9b), the separation along PC1 between fresh and longer-stored samples was mainly based on a wavelength at 1366 nm, while the separation along PC2 was due to 1385 and 1521 nm. According to Tsenkova et al. [24], the wavelengths at 1366 and 1385 nm correspond to the absorbance of water solvation shells.

The PLS regression between NIR spectra and I_LD_ data was also tentatively applied to the data from Trial 2. However, in this case, the results were not satisfactory. Better results were achieved after the application of Moving Average smoothing and the first derivative Norris Gap, obtaining R = 0.86 in calibration. However, the results in cross-validation were not as encouraging (R cross-val = 0.54), not making the model useful for practical use. It should be noted that the NIR spectra were acquired on both the sides of intact and sealed salad bags and, therefore, on the whole salad, including both leaves and petioles. Conversely, the I_LD_ data resulted from a destructive analysis carried out exclusively on the leaves. Furthermore, it can be observed from Figure 3b that the samples up to day 7 were characterized by a very low variability of I_LD_ (2.5–3.7), with a detrimental effect on quantitative calibrations [48]. These two aspects could explain the low performance obtained for this model. 

The aquagrams were built using the same WABS as in Trial 1 (Figure 10).

The aquagrams of Trial 2 showed a different and opposite profile compared to Trial 1. As previously reported, the presence of plant material with different physiological and metabolic characteristics, due to the different growing and harvesting season [38,39,43,44], might have affected the hydration state of the stored product. Furthermore, morphological variability could have influenced the aquagram trends.

Treatments A and B showed similar profiles, with the fresh product (t0) profile very different from all the others. Consequently, it was difficult to identify a trend as a function of storage time. Fresh samples located on the right side of the chart, showed a prevalence of free water, while the stored samples were more characterized by bound water. The profiles of checkpoints 1–13, on the other hand, looked very similar, and almost overlapped in the region of 1441–1478 nm. This could indicate that the major changes in water occurred immediately after packaging. Major variations occurred at 1416–1428 nm and between 1490–1509 nm. Conversely, treatment C showed different profiles for the various sampling days.

These results suggested building a dedicated series of aquagrams not only as a function of the applied perturbation but also of the variability of the raw material.

This topic should be investigated more in-depth to determine the role of water and interactions with plant tissues during storage.

## 3. Materials and Methods

Two experimentations were carried out at the Research Centre for Engineering and Agro-Food Processing of the Council for Agricultural Research and Economics (CREA IT, Milan, Italy). Freshly harvested rocket leaves (*Eruca sativa* Mill.) were purchased from a local grower in Milan (Italy). The first test was carried out in spring on the first cut rocket (Trial 1), while for the second experiment second cut autumn rocket (Trial 2) was used. The fresh salad was immediately transported to the CREA-IT lab and inspected for impurity and visual defects, discarding damaged and dehydrated leaves. The product underwent washing in mildly chlorinated water (1% sodium hypochlorite solution) for 1 min, followed by rinsing in cold water, and centrifugation to remove the remaining water. Rocket leaves (ca. 60 g) were packed in polypropylene film bags, and each bag was sealed under atmospheric and modified headspace conditions. The following gas mixtures were used: treatment A, atmospheric air (21% O_2_, 78% N_2_); treatment B, 30% O_2_, 70% N_2_; treatment C, 10% CO_2_, 5% O_2_, 85% N_2_. Three bags were prepared for each evaluation day for each treatment. All samples were stored at 4 ± 1 °C for 13 days and subjected to NIR, E-nose, and electrolyte leakage analyses at scheduled sampling points: 0, 1, 4, 7, 11, and 13 days for NIR and electrolyte leakage analyses; 1, 4, 7, 11, and 13 days for E-nose analyses.

### 3.1. E-Nose Analyses

The E-nose measurements were performed in triplicate on 3 bags for each treatment through a pierceable Silicon/Teflon disk, using a commercial portable electronic nose (PEN3, Win Muster Airsense Analytic Inc., Schwerim, Germany). The instrument is made up of a sampling apparatus, a detector unit containing the sensor array, and pattern-recognition software (Win Muster v.16) for data recording. The sensor array is composed of 10 metal oxide semiconductor (MOS) type chemical sensors: W1C (aromatic compounds), W5S (broad range), W3C (aromatic compounds), W6S (hydrogen), W5C (alkanes, aromatic compounds, less polar compounds), W1S (broad range, methane), W1W (terpenes, organic sulfur compounds, limonene, pyrazine), W2S (alcohols, broad range), W2W (aromatic compounds, organic sulfur compounds), and W3S (methane aliphatic, reacts on high concentrations > 100 mg/kg). The sensor response is given by the ratio between the conductivity response of the sensors to both the sample gas (G) and the carrier gas (G_0_) over time (G/G_0_). The E-nose analyses were performed following the conditions reported by Vanoli et al. [49]. For each measurement, the conductivity G/G_0_ of the 10 sensors at the time corresponding to the normalized maximum of all signals was taken as the vector of the sensors’ signal. The average of the measurements of each replicate was used for statistical analysis. Data were subjected to one-way analysis of variance (ANOVA) for means comparison using Statgraphics ver. 5.1 (Manugistic Inc, Rockville, MD, USA) software package. The means were separated using a Tukey’s HSD test, and their statistical significance was determined at 5% (*p* < 0.05) level. The E-nose data were also analyzed by principal component analysis (PCA) with the Unscrambler software package (v 9.7, Camo, Inondhcim, Norway).

### 3.2. Electrolyte Leakage Test

The index of leaf damage (I_LD_) was measured by an electrolyte leakage test. The analysis was performed in triplicate on 3 bags for each treatment. A method modified from Kim et al. [33] was used for ion leakage evaluation. Rocket leaf blades were cut into 1 cm^2^ pieces, washed in distilled water, and blotted dry with paper towels. Two grams of the cut samples were placed in a falcon tube containing 25 mL of Milli-Q water and shaken for 1 h in a horizontal shaker at 100 strokes/minute. The initial conductivity (IC) of the solution was measured with a digital conductivity meter (Accumet AR20, Fisher Scientific, Waltham, MA, USA), and the tubes were frozen at −20 °C for one week. Frozen samples were then held at room temperature for 24 h, shaken for 1 h, and final conductivity (FC) was measured. The index of leaf damage was calculated as follows: I_LD_ = (IC/FC) ∗ 100. Differences among averages were calculated by one-way ANOVA (Statgraphics 5.1 statistic software) and a least significant difference (LSD) post-hoc test. All the averages whose difference exceeds the LSD value are statistically significant (*p* < 0.05).

### 3.3. NIR Spectroscopy

Near-infrared spectra were collected in reflectance mode using the MicroNIR OnSite-W (VIAVI Solutions Italia S.r.l., Monza, Italy) portable spectrometer. Spectra were acquired in the spectral range between 900 and 1600 nm (50 scans; 125 reading points) on 3 bags for each treatment, on both sides of the bags. Ten replicates for each side were collected, for a total of 60 spectra for each treatment. Spectra of PET packaging (#10) were also collected, and the average spectrum was subtracted from each sample spectrum. Before the analysis, the instrument underwent calibration for the black, on the air, and for the white, on the supplied standard tile.

### 3.4. Chemometrics and Aquaphotomics

Chemometric analysis of the NIR data was performed using the Unscrambler software package (v 9.7, Camo, Inondhcim, Norway). 

An explorative PCA was applied on spectra pre-processed using the Moving Average smoothing (gap size 15 points) and the first derivative Norris Gap transformation (gap size 21 points) in the range of the first overtone of water (1300–1600 nm).

Predictive NIR models for the I_LD_ index were computed by partial least square (PLS) regression on spectra pretreated with the Standard Normal Variate (SNV) and the second derivative Savitzky–Golay transformation (gap size 11 points).

Spectra were also pretreated according to Tsenkova et al. [24]: the second derivative Savitzky–Golay filter (second-order polynomial fit and 21 points) and Multiplicative Scatter Correction were applied to absorbance spectra to remove potential scatter effects. The transformed spectra were then normalized by applying the following formula:(Aλ − μλ)/σλ(1)
where Aλ is the transformed absorbance, μλ is the mean value of all spectra and σλ is the standard deviation of all spectra at wavelength λ. The normalized spectra were then properly averaged using MS Excel^®^.

Twelve wavelengths that showed the maximum or minimum values in the normalized spectrum and the most impacting PCA or PLS loadings were chosen. The aquagrams were then built up using the selected wavelengths. 

## 4. Conclusions

E-nose and I_LD_ results agreed in identifying the B atmosphere as the best for maintaining the freshness of RTE rocket salad in both experiments.

According to E-nose results, the A and B atmospheres confirmed the plausible positive and active role of oxygen concentration in maintaining the freshness of ready-to-eat rocket and minimizing the occurrence of anaerobic fermentations. Treatment C with low oxygen concentration seemed to lead to a switch to anaerobic respiration with a release of alcohols.

Similarly, the index of leaf damage suggested that in the C atmosphere composition, the cell membrane underwent more degradation and damage than in the other treatments.

NIR spectroscopy was useful in observing samples distribution according to the storage time, mainly based on wavelengths corresponding to the absorbance of the water solvation shells. The aquagram proved to be a promising tool for detecting changes in the water structure during storage in atmospheres with different gaseous compositions. For Trial 1 the spider-charts gave rise to different WASPs for each sampling time and treatment, suggesting that treatment B was the best at preserving the freshness of rocket salad, according to the E-nose and I_LD_ results. For Trial 2, aquagrams could not provide a clear interpretation of the changes that occurred inside the bags. However, it was possible to identify a greater variability in the WASPs in the C treatment during the shelf-life, suggesting greater changes in the water structure compared to the A and B atmospheres. 

Although the data are very preliminary, the good agreement between the NIR spectra and the I_LD_ data makes NIR spectroscopy a promising technique for non-destructively evaluating the damage status of the plant membrane of the RTE rocket salad during the shelf-life.

NIR coupled with Aquaphotomics should be investigated more in-depth to determine the role of water and interactions with plant tissues during shelf-life in order to confirm these preliminary results.

Some other destructive parameters, such as consistency, fluorescence, ripening index, are under evaluation to confirm these results obtained, using mainly rapid and non-destructive techniques.

## Figures and Tables

**Figure 1 molecules-27-02252-f001:**
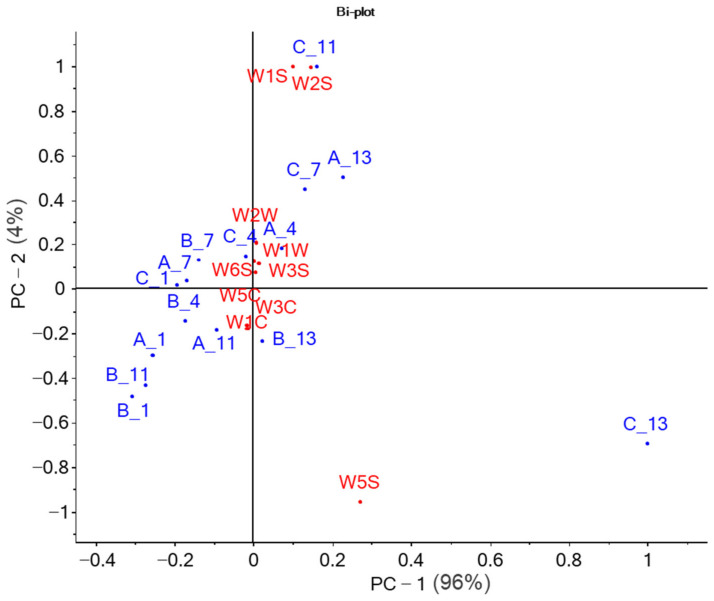
Biplot of E-nose sensor responses and samples scores (average values) during the storage of rocket salad.

**Figure 2 molecules-27-02252-f002:**
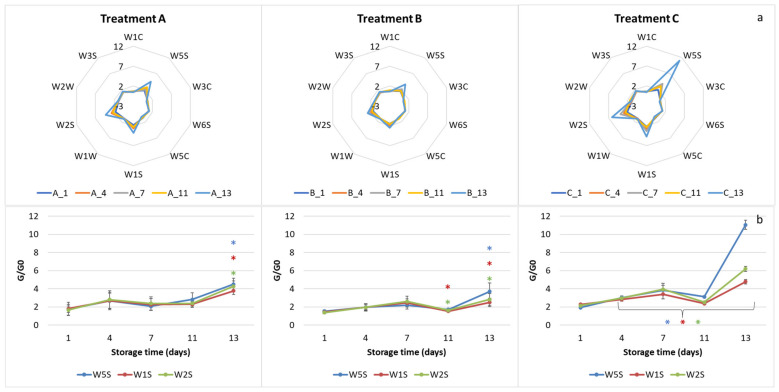
E- nose profiles (**a**) and trend curves of W5S, W1S and W2S sensors (**b**) of the three treatments during the storage: Trial 2. Data are reported as mean values ± standard deviation. Colored asterisks indicate significant differences in the corresponding sensor signals according to ANOVA and Tukey’s post hoc test (*p* < 0.05).

**Figure 5 molecules-27-02252-f005:**
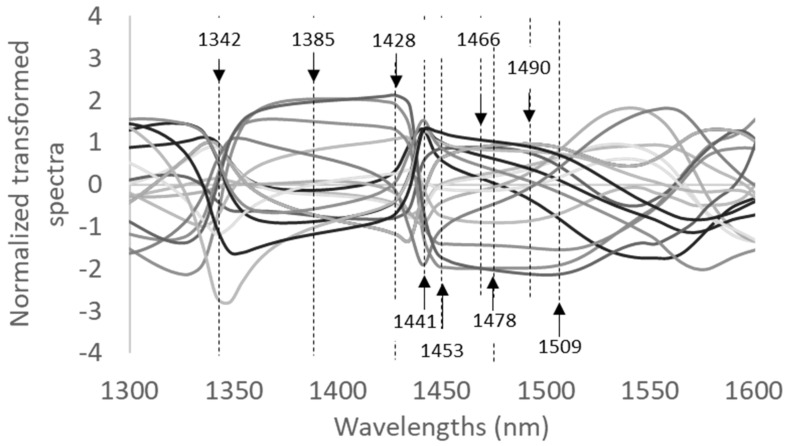
Normalized spectra of the packed rocket salad samples of Trial 1.

**Figure 6 molecules-27-02252-f006:**
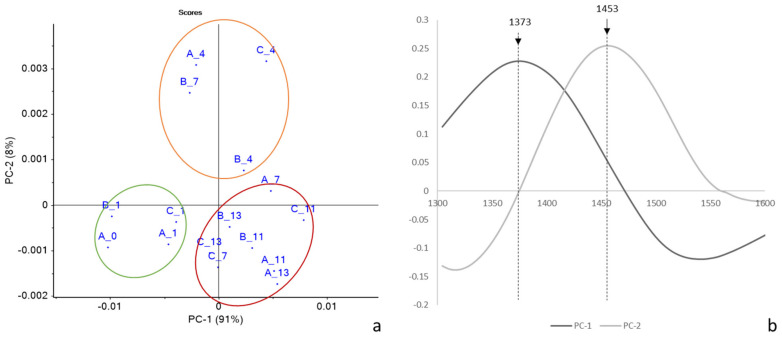
PCA score (**a**) and loading (**b**) plots of pre-treated NIR spectra of rocket salads of Trial 1 (each point corresponded to the mean spectra for each checkpoint).

**Figure 7 molecules-27-02252-f007:**
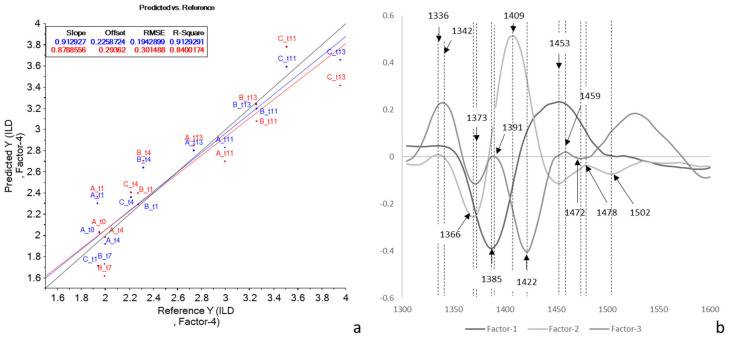
Scatter plot of calibration (blue) and cross validation (red)performance of predictive models built with pretreated NIR spectra and I_LD_ values (**a**); PLS Factor Loadings (**b**).

**Figure 8 molecules-27-02252-f008:**
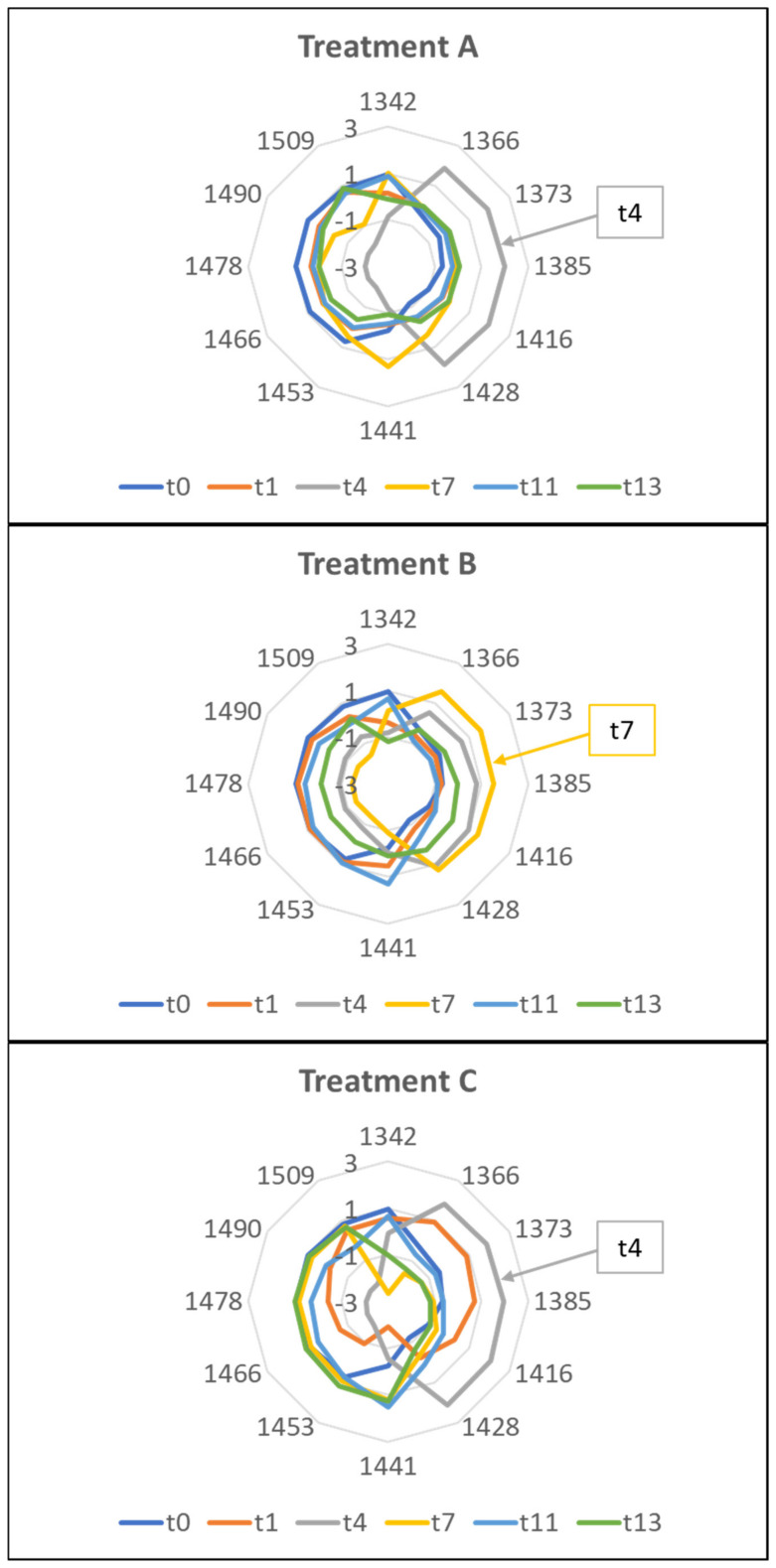
Aquagrams of the rocket samples belonging to the three treatments during the storage: Trial 1.

**Figure 9 molecules-27-02252-f009:**
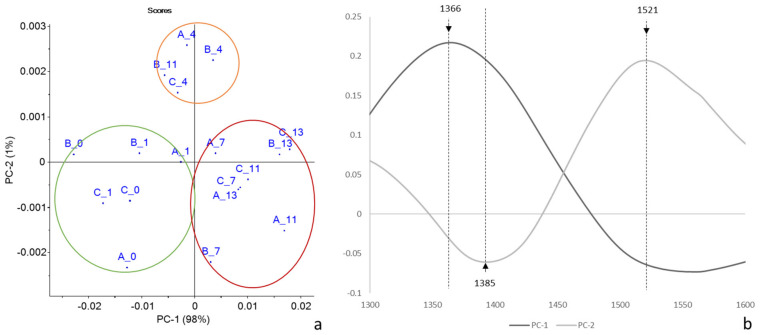
PCA score (**a**) and loading (**b**) plots of pre-treated NIR spectra of rocket salads of Trial 2 (each point corresponded to the mean spectra for each checkpoint).

**Figure 10 molecules-27-02252-f010:**
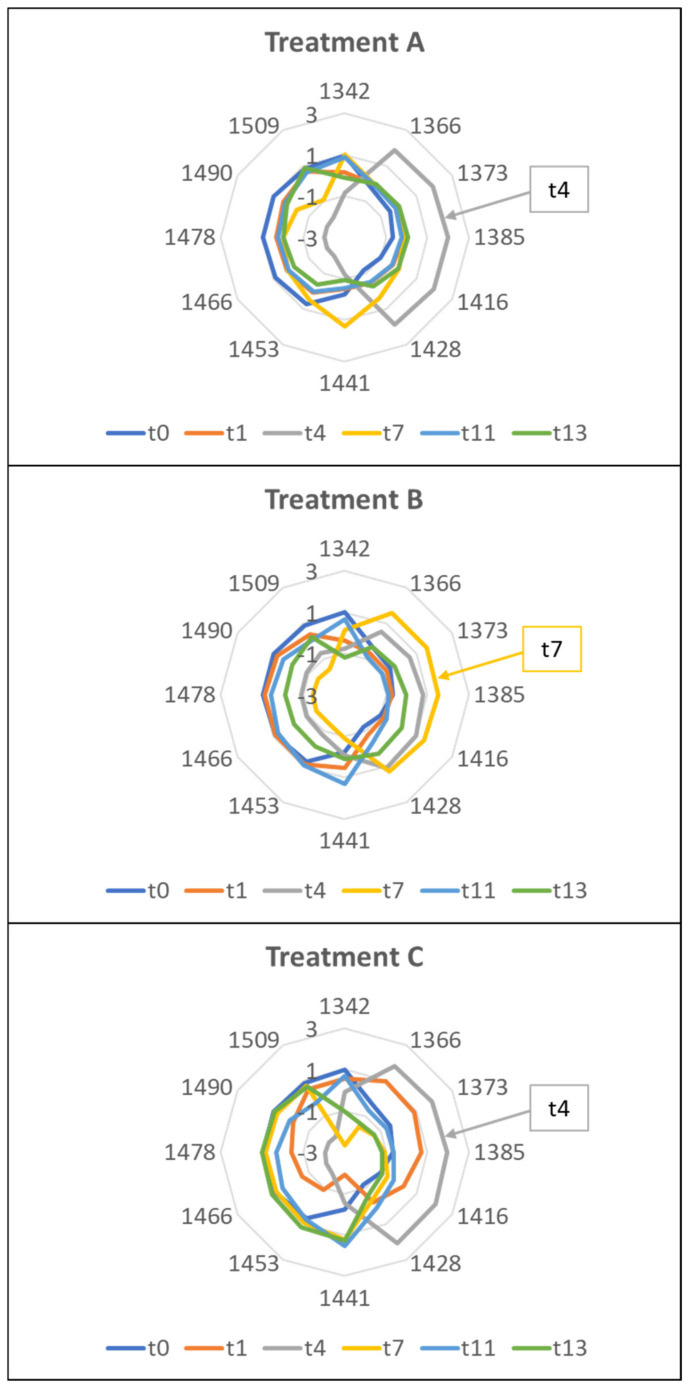
Aquagrams of the rocket samples belonging to the three treatments during the storage: Trial 2.

**Table 1 molecules-27-02252-t001:** Selected wavelengths.

Selected Wavelengths	WAMACs [22]
1342	C1—ν3
1366	C2—water solvation shell, OH-(H_2_O) n, n = 1, 2, 4
1373	C3—ν1 + ν3
1385	C4—water solvation shell, OH-(H_2_O)1,4 and superoxide, O2-(H_2_O)4
1416	C5—free water molecules (S0)
1428	C6—water hydration, H-OH bend and O…O
1441	C7—water molecules with 1 hydrogen bond (S1)
1453	C8—ν2 + ν3, Water solvation shell, OH-(H_2_O)4,5
1466	C9—water molecules with 2 hydrogen bonds (S2)
1478	C10—water molecules with 3 hydrogen bonds (S3)
1490	C11—water molecules with 4 hydrogen bonds (S4)
1509	C12—ν1, ν2, strongly bound water

## Data Availability

The data presented in this study are not publicly available due to the privacy related to the Agridigit project.

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
