# Peer review of "Aquaphotomic, E-Nose and Electrolyte Leakage to Monitor Quality Changes during the Storage of Ready-to-Eat Rocket"

_molecules, 2022, doi:10.3390/molecules27072252_

Round 1

Reviewer 1 Report

The manuscript number molecules-1614358 aimed to evaluate the feasibility of different analytical (non-invasive) techniques (e.g., aquaphotomic, E-nose and electrolyte leakage) to infer about the shelf-life of ready-to-eat rocket.

The topic is interesting and in principle it has a practical and relevant application.

However, in my opinion there are several flaws that do not allow to consider it for publication.

1) Throughout the text, it is not demonstrated for which storage time (if any) the quality of the ready-to-eat rocket samples were compromised or if during the 13 days of analysis, the rocket samples could be consumed or not.

From the text, with the possible exception of the data regarding the index of leaf damage, no other quantitative data was assessed that could be used as an indicator of shelf-life or if the rocket samples were or not proper for consumption.

Thus, other standard evaluation should be carried out (namely regarding the microorganisms loads, the perception of sensory defects, or the changes in volatiles composition) that could be related and/or used to know the product suffered any chemical-sensory degradation that could compromise its consumption and then, if the proposed techniques could be applied to check the process.

2) From the text it is not obvious why gas mixtures A and B would be so superior compared to gas mixture C. The conclusion is only based on the index of leaf damage, and other parameters are more important than that, for sure. Besides, it is never said what in an acceptable (or not) index. Which value is usually assumed as a negative threshold?

3) From the text, and contrary to what the Authors try to imply, none of the three techniques applied (aquaphotomic, NIR, E-nose) can really split the samples according to the gas mixture treatment/storage time-period. In Figures 1, 6 and 9, it is clear that different gas mixtures and time-periods are "grouped" into the same clusters.

Also, it should be explained in Figures 6 and 9, how the circles/ellipses were obtained. Are they drawn or computed based on a statistical analysis? This is crucial to realize if the proposed grouping is based on not in any scientific/statistical evidence. 

For example, in Figure 6, sample B_4 is clustered with B_7, A_4 and C_4 although it is closer to A_7 samples?

For example, from Figure 9, it is said that "The group of samples from day 4 was separated from the others along PC2". However, this is not totally true since the cluster also includes B_11 samples...

4) Finally, what we can observe from the data is that the results would be seasonally dependent, with opposite findings arising from trials 1 and 2, and so, no possible extrapolation could be foreseen. Thus, the useful of the study is low.

Reviewer 2 Report

This manuscript was well written, all figures in this manuscript were
perfect. But there are some problems to English grammar, the author
should revised carefully.

1).Could the author explain the important role of putrescine in the plant metabolite and increase the paragraph in the introduction of revised manuscript.

2).What is the relationship between Polyamines (PAs) and putrescine?

3).What is the importance of putrescine to plant growth and biotic stress?

4).Are there have any recent research related to the putrescine and plant growth or metabolite?

5).The references used in this study was old, the author should find somes research related to this topic in the year of 2019-2022.

Reviewer 3 Report

Manuscript molecules-1614358 reportes on the use of non-destructive methods, such as Aquatophotomics, NIR and E-nose for the shelf life prdiction of rocket salad. There are limited studies avaialable using this hypothesis and given the fact that fruit and vegetables aid to a good health balance, the study is vovel in the field. Data are convincing and have been traeted properly with chemometrics.

As far as the English language and technical quality, these are both professionally. I have indicated minor suggestions for authors within the attached pdf to improve a littele bit more their work.

Based on these comments, I suggest a minor rvision prior to further consideration for publication.

Reviewer 4 Report

Review of manuscript number Molecules-1614358 

Title: E-nose and electrolyte leakage for the evaluation of the shelf-life of ready-to-eat rocket

The article describes the results of the investigation on RTE rocket salad freshness, which were packaged under different conditions (atmospheric air, 30% of oxygen, 5% of CO2, and 80% of N2). The authors performed two trials of the same experiment but the first was carried out in spring and the second in autumn. The analysis carried out in this study were E-nose, NIR spectroscopy, and index of leaves damage measured by electrolyte leak. All investigations were performed after 0,1,4,7,11 and 13 storage at 4°C. Some parts of the paper should have been changed.

Keywords: Too many words were provided, according to instruction for authors …” Three to ten pertinent keywords need to be added after the abstract”...

Introduction

Line 67- 69 Please add literature to this paragraph. Could authors give some example of volatile that can be named as a marker of such metabolic changes?

Material and methods

Can authors explain in which country, the region the plants were harvested?

I think that this paragraph should be divided into a few parts called e.g. E-nose, Index of leaf damage, Near-infrared analysis, statistical analysis. Can the author explain how many replications of each analysis were performed?

Results and discussion

E-nose analysis

Can the authors explain any difference between  A and C samples in trial 1? does the atmosphere of the package change the aroma profile of samples A and B?

Line 151 – 152 Can authors prove that in sample C the release of alcohols was significant? Can these components change the quality of the product?

Can the author explain how long the product should be stored to maintain the best sensory quality?

2.2 Electrolyte leakage

Line 181-182 can the authors add some other data that proves the thesis of the impact of temperature and rainfall on the leaves composition?

2.3. NIR spectroscopy and aquaphotomics

Line 214 – 215 Please explain what kind of pre-treatment of spectra was applied?

Conclusion:

This section is too long, please rewrite it. Some parts of this paragraph are repeated from lines 333-334. Please change the conclusion in relation to the aim of the study, and the show is that NIR spectroscopy and aquaphotomics are adequate methods for this kind of analysis.

Round 2

Reviewer 1 Report

The revised version of the manuscript made an effort to overcome the main concerns of the Reviewer, listed in the first report.

Although the effort has been made, in my opinion, the answers provided could not overcome the initial concerns, neither clearly show the practical application of the proposed methodologies, namely without supporting the findings in a comparison with new experimental data (chromoographic data and/or microbiological evaluations, or other) that could confirm the quality degradation of the rocket samples, only based on the index of leaf damage.

Author Response

This is an in-depth study to confirm the applicability of aquaphotomics in the food sector in order to improve the WAMACS matrix with additional biosystems that can be indirectly identified by exploiting water absorption.
This work wants to be the starting point to verify the possibility of selecting a few wavelengths, associated with the absorption of water, suitable for describing a bioprocess and constructing calibrations (with instrumental chemical data) for the prediction of the shelf-life or the quality decay with low-cost NIR tools.

The Authors agree with the Guest Editors that for this study no additional experimental data are needed.

Reviewer 2 Report

The author had corrected the manuscript according to my comments.

Author Response

The Authors thank for the approval of the revisions.

Reviewer 4 Report

Most of my comments was taken into the consideration.

Author Response

(The authors gave the same response as above.)
